# Firing of Replication Origins Is Disturbed by a CDK4/6 Inhibitor in a pRb-Independent Manner

**DOI:** 10.3390/ijms241310629

**Published:** 2023-06-25

**Authors:** Su-Jung Kim, Chrystelle Maric, Lina-Marie Briu, Fabien Fauchereau, Giuseppe Baldacci, Michelle Debatisse, Stéphane Koundrioukoff, Jean-Charles Cadoret

**Affiliations:** 1CNRS, Institut Jacques Monod, Université Paris Cité, F-75013 Paris, France; 2CNRS UMR9019, Institut Gustave Roussy, 94805 Villejuif, France; 3Sorbonne Université, 75005 Paris, France

**Keywords:** replicative stress, replication origins, CDK4/6 inhibitors

## Abstract

Over the last decade, CDK4/6 inhibitors (palbociclib, ribociclib and abemaciclib) have emerged as promising anticancer drugs. Numerous studies have demonstrated that CDK4/6 inhibitors efficiently block the pRb-E2F pathway and induce cell cycle arrest in pRb-proficient cells. Based on these studies, the inhibitors have been approved by the FDA for treatment of advanced hormonal receptor (HR) positive breast cancers in combination with hormonal therapy. However, some evidence has recently shown unexpected effects of the inhibitors, underlining a need to characterize the effects of CDK4/6 inhibitors beyond pRb. Our study demonstrates how palbociclib impairs origin firing in the DNA replication process in pRb-deficient cell lines. Strikingly, despite the absence of pRb, cells treated with palbociclib synthesize less DNA while showing no cell cycle arrest. Furthermore, this CDK4/6 inhibitor treatment disturbs the temporal program of DNA replication and reduces the density of replication forks. Cells treated with palbociclib show a defect in the loading of the Pre-initiation complex (Pre-IC) proteins on chromatin, indicating a reduced initiation of DNA replication. Our findings highlight hidden effects of palbociclib on the dynamics of DNA replication and of its cytotoxic consequences on cell viability in the absence of pRb. This study provides a potential therapeutic application of palbociclib in combination with other drugs to target genomic instability in pRB-deficient cancers.

## 1. Introduction

Palbociclib is a small molecule that is part of the third generation of Cyclin Dependent Kinase (CDK) inhibitors, along with ribociclib and abemaciclib. These three drugs are highly selective and specific CDK4/6 inhibitors showing IC50 values at the nanomolar range in vitro [1]. These inhibitors showed their efficacy and safety in many clinical trials [2]. Based on successful outcomes, the United States Food and Drug Administration (FDA) approved palbociclib as a treatment for advanced breast cancer in 2015 (https://www.fda.gov/drugs/resources-information-approved-drugs/palbociclib-ibrance, accessed on 31 March 2017) [3]. Today, there are more than 200 ongoing clinical trials for different types of cancer, indicating palbociclib as a promising cancer treatment [4].

In an attempt to understand the antiproliferative molecular mechanism of palbociclib, numerous preclinical studies have investigated how palbociclib impacts downstream pathways of CDK4/6 in various cancer cell lines [5,6]. The most studied pathway is mediated by the retinoblastoma protein pRb (retinoblastoma protein), which is the major substrate of CDK4/6. Indeed, pRb plays pivotal roles in the G1/S transition of the cell cycle, with its phosphorylated form releasing the E2F family of transcription factors, which in turn regulate the expression of genes involved in the progression of the cell cycle and in DNA replication [7,8]. CDK4/6 participates in this cascade as kinases phosphorylate the Serine 780/795 and Serine 807/811 of pRb. Upon treatment with palbociclib in clinical doses (around 1 μM), pRb remains underphosphorylated and sequesters E2F factors, which finally induces efficient cell cycle arrest in G1 [9]. However, emerging evidence has shown that palbociclib has additional effects at high concentrations in a CDK4/6 or pRb-independent manner [10,11,12,13]. To emphasize these findings, we reveal here the pRb-independent actions of palbociclib on the dynamics of DNA replication at high doses.

DNA replication is a highly regulated process that ensures the duplication of genetic materials to transmit the entire genome to daughter cells. In human cells, thousands of origins are licensed through the late M and early G1 phase, requiring the loading of the pre-replication complex (pre-RC) onto chromatin. Then, only a subset of licensed origins fires during the S phase [14,15]. Increasing activity of S-phase related CDKs and DDK (Dbf4-dependent kinase) ensures the activation of origins at the transition from the pre-RC to the pre-initiation complex (pre-IC). Indeed, DDK, a complex of Cdc7 with the Dbf4 catalytic subunit, is essential for the initiation of DNA replication [16,17]. In collaboration with S-phase CDKs, DDK recruits the helicase activators Cdc45 (Cell division cycle 45) and GINS (Go-Ichi-Ni-San) on chromatin, leading to the formation of the active replicative helicase CMG (Cdc45-Mcm-GINS) complex [18]. Defects in origin efficiency could induce replicative stress and DNA damage and threaten the accuracy and completion of DNA replication, which may lead to genomic instability [19,20].

In this study, we focused on two pRb-deficient cancer cell lines MDA-MB-468 and NCI-H295R, which are derived from breast cancer and adrenocortical carcinoma, respectively. We demonstrate novel actions of palbociclib impairing the initiation of DNA replication. Indeed, we show that palbociclib induces an impairment of DNA synthesis during the S phase of the cell cycle. By using different molecular approaches, we precisely decipher how and at which level palbociclib perturbs DNA replication and the genomic stability in a pRb-negative context. Thus, this study brings palbociclib into focus as a potential treatment targeting replicative stress for patients suffering from pRb-deficient cancers.

## 2. Results

### 2.1. Palbociclib Affects Cell Viability of pRb-Deficient Cell Lines

To study the effect of CDK4/6 inhibitors on cellular viability, two pRb-deficient cell lines, namely MDA-MB-468 and NCI-H295R, were treated with increasing doses of either palbociclib or ribociclib. Palbociclib reduced cellular viability in a dose-dependent manner while ribociclib had less effect on viability in both cell lines at equivalent concentrations, indicating that palbociclib has additional cytotoxic effects in a pRb-negative context (Figure 1A,B). In order to explore the nature of the impact on cellular viability, we performed an apoptosis analysis based on the activity of caspase3/7. MDA-MB-468 cells treated with 15 or 20 μM of palbociclib showed a high activity of caspase3/7, whereas ribociclib had a modest effect below 20 μM (Figure 1C). NCI-H295R cells went into apoptosis at 15 μM palbociclib, and the caspase activity could not be observed at 20 μM due to massive cell death (Figure 1D). This finding shows that relatively high doses of palbociclib might have other cellular effects that do not require the presence of pRb, as observed in former studies [6]. For all further experiments, we treated cells with 10 µM of palbociclib or ribociclib, as this concentration of palbociclib significantly reduced cell viability without inducing cell death by apoptosis. Of note, cells were treated for approximately one doubling time (48 h for MDA-MB-468 cells and 96 h for NCI-H295R).

As a second step, the effects of palbociclib and ribociclib on the cell cycle were investigated. Treatment with palbociclib modestly affected the cell cycle distribution, with only a slight increase in the proportion of cells in G1 and G2/M phases in MDA-MB-468 and in G2/M in NCI-H295R cells (Appendix A). Considering the duration of treatment (one doubling time), our results suggest that there may be an extension of the G1 or G2 with no cell cycle arrest in both cell lines.

### 2.2. Palbociclib Treatment Impairs DNA Synthesis during the S Phase in pRb Deficient Cells

We further analyzed the cell cycle by monitoring the EdU incorporation assay to characterize how cells behave during the S phase of the cell cycle. This analysis allowed us to visualize the amount of newly synthesized DNA in both cell lines (Figure 2A,B). Strikingly, cells treated with palbociclib showed a lower intensity of EdU compared to mock treatment, whereas ribociclib had minor effect on EdU incorporation (Figure 2C). Comparing medians of EdU incorporation in both cell lines, we noticed that MDA-MB-468 cells incorporate more EdU than NCI-H295R cells consistently with the length of their respective doubling time (Appendix A). Palbociclib reduced the level of EdU incorporation in MDA-MB-468 cells and NCI-H295R cells by 40% and 50%, respectively (Figure 2C). Furthermore, the decrease of EdU incorporation in NCI-H295R cells treated with palbociclib was similar to the level in cells treated with 50 or 100 µM hydroxyurea (HU), which are considered mild replicative stress (Appendix A). These results suggest that palbociclib efficiently impairs the capacity of cells to synthesize DNA during the S phase, thus implying that palbociclib may interfere with DNA replication. Since we observed that palbociclib affects DNA synthesis, we investigated the impact of palbociclib on the presence of replicative stress and DNA damage. The activation of S-phase checkpoint was measured by the Serine 345 phosphorylation status of the checkpoint Serine/Threonine kinase 1 (Chk1). We observed a two-fold increased level of phosphorylated Chk1 protein with palbociclib treatment (2.41-fold in MDA-MB-468 and 2.13-fold in NCI-H295R cells), while the total amount of Chk1 remained constant in both cell lines (Figure 2D,E). However, ribociclib showed moderate effects on the Chk1 phosphorylation status in MDA-MB-468 cells and no effects on NCI-H295R cells. The phosphorylation level of Chk1 after treatment with palbociclib was similar to mild replicative stress induced by HU treatment. Then, we observed the induction level of DNA damage measured via γH2AX levels. We demonstrated that palbociclib triggered the accumulation of γH2AX phosphoprotein, whereas ribociclib had minimal effects on both cell lines. We concluded that palbociclib treatment drove cells to accumulate more double strand breaks in DNA. This could partially explain the modest increase of cells in the G2 phase that was previously observed (Appendix A). Cells may require a longer duration of the G2 phase to manage a complete DNA repair. Hence, cells treated with palbociclib exhibit the activation of the S-phase checkpoint and a sign of genomic instability along with interferences in DNA synthesis during the DNA replication process.

### 2.3. Palbociclib Alters DNA Replication Timing

In order to determine how palbociclib affects the DNA replication process on a genome-wide scale, we performed an analysis of the temporal program of DNA replication, as previously described [21]. Cells organize DNA replication in space and time through the S phase. This spatial and temporal program of replication orchestrates a faithful duplication of the genetic materials during the S phase of the cell cycle. Analysis of the temporal program allows a characterization of the effects of replicative stress and a visualization of the regions of the genome that are altered in terms of replication timing. To perform this technique, the cells are labeled with a short pulse of BrdU and separated into two fractions corresponding to early and late S phase. The ratio between these two fractions is calculated and visualized for the entire genome. A comparison of the profiles of the control and treated cells revealed that the palbociclib induced perturbations of the replication timing in several parts of the genome (Figure 3A,C), while the ribociclib treatment induced minimal effect (Figure 3B,D). In the MDA-MB-468 cell line, 16.82% of the genome showed an altered replication timing program after the palbociclib treatment when compared to the mock-treated control cells (Figure 3E). Among the regions harboring altered replication timing, 62.5% displayed a delayed replication timing, whereas 37.5% showed an advanced one (Figure 3F). In the NCI-H295R cell line, 4.12% of the replication timing program was altered after the same treatment, with 77.1% and 22.9% of delayed and advanced replicating regions, respectively. These results indicate that palbociclib perturbs the timing of DNA replication at the genome scale (Appendix A).

### 2.4. Palbociclib Reduces the Number of Active Origins without Altering Replication Fork Progression

To decipher how palbociclib impacts DNA replication dynamics, we performed DNA fiber analysis to measure the speed and density of replication forks and the density of activated replication origins [22]. We first examined the fork speed after a treatment with palbociclib in the MDA-MB-468 cells. In the control cells, the median fork speed was 2.01 kb/min (Figure 4A,B) and showed a slight increase (2.16 kb/min) after the palbociclib treatment (although not significant, *p*-value = 0.068, *t*-test). The NCI-H295R cells presented a relatively low fork speed (1.18 kb/min) in the control condition but a statistically significant increase to 1.47 kb/min after the treatment (*p*-value = 0.028, *t*-test). We observed no significant difference in the replication fork asymmetry, indicating that the elongation step is not altered in both cell lines (Figure 4C). Next, we investigated whether palbociclib could impair the activation of replication origins by measuring the density of forks. After the treatment with palbociclib, the fork density was reduced by 35% and 40% in the MDA-MB-468 and NCI-H295R cells, respectively (Figure 4D). It has been observed that replicative stress can induce concomitant effects on origin activation and fork speed, which both influence each other [23]. Regarding our data, reduced origin firing could trigger a slight acceleration of fork speed that compensates for this lack of activated origins. Our results show that palbociclib reduces the number of active origins but does not interfere with replication fork progression, which provides a mechanistic basis for the reduced EdU incorporation observed in S-phase cells via flow cytometry (Figure 2A,B).

### 2.5. Palbociclib Acts on the Quantity of Pre-Initiation Complex Proteins

To better understand the molecular process affected by palbociclib, we investigated the DNA replication initiation process. We first looked at the chromatin loading of proteins constituting the Pre-Replication Complex (Pre-RC) namely Orc1, Cdt1 and Mcm2. In both MDA-MB-468 and NCI-H295R cell lines, the levels of these proteins remain constant (Figure 5A,B) in the different experimental conditions. We further checked the phosphorylation level of Mcm2 and the loading on chromatin of Cdc7, Dbf4 and Cdc45 that are involved in the formation of pre-initiation complex (Pre-IC) (Figure 5C). We found that the loaded amount of Cdc7 and Cdc45 and the level of phosphorylation of Mcm2 were decreased in MDA-MB-468 cells treated with palbociclib (Figure 5D). In NCI-H295R cells, palbociclib treatment efficiently decreased the loaded amount of Cdc7, Dbf4 and Cdc45 (Figure 5E). These results indicate that palbociclib does not interfere with the Pre-RC loading but impairs the assembly of the pre-IC onto chromatin, resulting in reduced origin firing. We further studied whether the decrease of chromatin-bound proteins was due to a variation in the total quantity of proteins or to a defect in the pre-IC loading process. Actually, the total amounts of Cdc7 and Cdc45 pre-IC proteins decreased in MDA-MB-468 cells treated with palbociclib (Figure 5F,G), which is consistent with the results obtained with chromatin extracts. At the same time, lower amounts of Cdc7, Dbf4 and Cdc45 proteins were also detected in palbociclib-treated NCI-H295R cells (Figure 5F,H).

### 2.6. Palbociclib Affects the Expression of Genes Coding Pre-Initiation Complex Proteins

Then, we performed RT-qPCR to check whether palbociclib affects mRNA levels of genes encoding pre-IC proteins. We detected significantly reduced levels of the expression of *Cdc7* and *Cdc45* transcripts in MDA-MB-468 cells after 24 h of treatment (Figure 5I). In NCI-H295R cells, the expression levels of *Cdc7* and *Cdc45* were both reduced after 48 h of treatment with palbociclib (Figure 5J). Consistent with proteins level analysis, *Cdc45* and *Cdc7* are repressed at the transcriptional level in both cell lines. However, we detected no significant variation in the level of *Dbf4* mRNA whereas the Dbf4 protein is less present in NCI-H295R cells treated with palbociclib. This could be explained by either the stability of the protein due to post translational modifications (phosphorylation or ubiquitination) or to its binding with Cdc7 which could contribute to the change of Dbf4 stability. Furthermore, we detected no significant variations in the levels of *Orc1* and *Mcm2* mRNA in both cell lines after the treatment. Collectively, our results suggest that palbociclib impairs the expression level of specific genes particularly involved in the formation of the pre-IC in pRb-deficient cell lines. This explains how palbociclib impairs DNA synthesis through the partial dysfunction of origin activation due to lacking of Pre-IC components. We revealed that palbociclib actions are mediated by transcriptional regulations of genes coding pre-IC proteins.

## 3. Discussion

The clinical dose of palbociclib is usually around 1 µM for pRb+/+ cancers, such as breast cancer indicating that palbociclib is a promising and efficient cancer drug. In pRb-positive cancer, neutropenia seems to be the predominant effect of a palbociclib treatment with an associated tolerance depending on individuals. Today, palbociclib treatment is also associated with Endocrine Therapy. On these long-term safety analyses, there is no evidence of specific cumulative or delayed toxicities [24]. The main targets of this drug are CDK4/6 which will inhibit pRb phosphorylation and thus block E2F transcription factors that regulate many of the genes involved in the cell cycle. Since pRb is the major phosphorylation substrate of CDK4/6, pRb loss is largely considered as a biomarker for the resistance to CDK4/6 inhibitors [2,4]. In our study, we decided to use pRb-deficient cells to observe the effects of these drugs in non-canonical pathways, highlighting the hidden effects of palbociclib beyond the pRb pathway. In agreement with other studies, we detected little effect of palbociclib or ribociclib around 1 µM which is considered the specific inhibitory dose for pRb-proficient cell lines (Figure 1) [5,25]. Nonetheless, palbociclib significantly reduced the cell viability at 10 µM, indicating that this drug has cytotoxic effects in a pRb negative context. Since the effects on cellular viability are barely observed with this same concentration of ribociclib (Figure 1), we then suppose the actions of palbociclib are mediated by its off-targets. Recent multi-omic studies have demonstrated the different specificity of three CDK4/6 inhibitors (ribociclib, palbociclib and abemaciclib) focusing on a large spectrum of targets for abemaciclib and also revealed that palbociclib targets other kinases than CDK4 and CDK6 at high doses [1,13].

Moreover, we observed a decreased level in DNA synthesis after a treatment with palbociclib in pRb-deficient cell lines (Figure 2). The treatment of pRb-deficient cells does not trigger any cell cycle arrest, nor any variation in the proportion of S-phase cells (Appendix A), but we noticed reduced DNA replication efficiency during the S phase of the cell cycle (Figure 2). We also observed an accumulation of cells in G2 phase after these treatments. One hypothesis to explain this accumulation in G2 is to give the cells the opportunity to perform DNA damage repair and/or complete replication by the MIDAS process. We also demonstrated the effects of palbociclib on DNA replication at the whole genome scale by studying the temporal program of DNA replication. We observed that palbociclib disturbs the DNA replication timing, inducing significant delays at certain regions (Figure 3). Further, palbociclib treatment induces replicative stress (Figure 2D) that may be caused by either the inactivation of some replication origins, or by replicative fork arrest that induces the collapse of replisomes during the elongation stage [26].

To decipher the mechanism of action of palbociclib on the dynamics of DNA replication, we performed replication dynamic studies using the DNA fiber assay (Figure 4). Our findings allowed us to eliminate the hypothesis that palbociclib may affect the elongation of replication since no evidence has been found for fork slowing or collapse. However, the DNA combing assay indicates a reduction of the density of replicative forks by 35–40%. This decrease in the density of replication forks could be due to a defect in the initiation of origins. We then studied the loading onto the chromatin of the pre-RC, the first complex associated with the replication origins (Figure 5A,B) and observed that palbociclib treatment does not affect this loading. Since palbocilib treatment also affects several kinases, some of which could be involved in the phosphorylation of Pre-IC components, we wondered whether the activation of replication origins phase via the pre-IC phosphorylation was impacted (Figure 5C–E). We detected that the amounts of proteins constituting the pre-IC in chromatin extracts, such as Cdc7, Dbf4 and Cdc45 are reduced after the palbociclib treatment in the NCI-H295R cell line. The same is true for Cdc7 and Cdc45 in the MDA-MB-468 cell line. It is clear that the decrease of MCM2 phosphorylation observed (Figure 5C–E) is linked to the decrease of Cdc7 proteins, the CDK which activates the Pre-IC. The loading defect of Pre-IC on chromatin may also be due to a hindrance of loading of Cdc7 by the presence of p21 and/or to a decrease in the availability of Cdc45 proteins in the nucleus [27]). To answer this question, we looked at the level of these proteins in the total extract fraction (Figure 5F–H). For both cell lines, we observed a significant decrease in the amount of Cdc7 and Cdc45. We also observed a significant decrease of Dbf4 only in the NCI-H295R cell line. This observed decrease of these available proteins in each cell line is correlated with the reduction of the expression of the corresponding genes (Figure 5I,J).

How could palbociclib alter the expression of *Cdc7* and *Cdc45* genes in a pRB independent manner? The regulation of expression of numerous genes has been studied in pRb-deficient cell lines under a treatment with 1 µM palbociclib as a negative control of their expression in a pRb-efficient cell line. However, no conclusion was drawn concerning the detected gene expression variations in pRb-deficient cells [28,29]. In our study, we asked whether the 10-fold higher dose would induce a non-canonical pathway or induce the repression of the E2F pathway despite the absence of pRb. However, in a pRb−/− cell context, we know that palbociclib cannot interact with unphosphorylated CDK4 [30]. Then, palbociclib could interact with the CyclinE/CDK2 complex, an off-target, to inactivate this non-canonical pathway [11,31]. Multi-omic analysis have shown that palbociclib inhibits CDK2/cyclinA1 and CDK2/cylinE1 in the µM range [12]. However, a study revealed that 10 µM of palbociclib, the concentration of palbociclib that we used in pRb-deficient cells, has a target occupancy of 65% for CDK2 [11], making it sufficient to target CDK2.

However, palbociclib could preserve at least in part the expression of transcription factors controlling the expression of Cdc45 and Dbf4 as it is performed by the CDK4/6-pRb-E2F pathway for the expression of a number of genes related to the G1/S transition. The final effect is then an inhibition of the expression of part of the Pre-IC genes. We could validate this hypothesis using CDK2 inhibitors, but currently, none of them have been shown to be particularly specific since they also target other CDKs such as CDK1. This prevented us from testing this hypothesis.

Palbociclib is an efficient CDK4/6 inhibitor inducing cell cycle arrest at the G1-phase in pRb-proficient cells, which is not the case in our study using two pRb-deficient cell lines. However, pRb does not act exclusively in the E2F pathway but also in chromatin remodeling by interacting with histone deacetylases (HDACs) and the ATP-dependent SWI/SNF complex [32,33], which implies that pRb-deficient cells exhibit a higher intrinsic genomic instability [34,35,36]. In parallel, it has been well studied that a defect of origin firing could provoke an incomplete replication of the genome, referred to as under-replication, thus inducing replicative stress and generating genomic instability [37]. Indeed, the proteins of pre-IC are limiting factors involved in the initiation of replication and their deregulation is tightly linked to genomic instability and cancer development [38]. For example, the depletion of Cdc7 with RNAi reduced the activation of origins at the initiation step, inducing replicative stress in the cells [39,40]. Taken together, targeting pRb-deficient cancers harboring intrinsic genomic instability with palbociclib could be a powerful method to increase the cytotoxic effect of this treatment.

In conclusion, the model we propose explains the actions of palbociclib in inducing replication stress and DNA damage in a pRb-negative context. Since the loss of pRb is associated with higher vulnerability of the cells to replicative stress, palbociclib may potentially be used in cancer treatment to exacerbate the genomic instability and vulnerability of cancer cells that are more sensitive to replicative stress. A recent paper from Knudsen’s team shows that pRb-deficient cells, which are resistant to specific chemotherapeutic agents such as carboplatin, are highly sensitive to PLK1 and CHK1 inhibitors. These inhibitors reduce pRb-deficient cells survival [41]. The same study also shows a significant effect on pRb-deficient xenografts. Finally, the authors suggest that this type of inhibitor could be used in pRb-deficient triple-negative breast cancer. From this point of view, we suggest that palbociclib may be used for pRb-deficient tumors combined with other therapeutic molecules inhibiting PLK1 and CHK1, such as Volasertib and AZD7762, respectively.

## 4. Materials and Methods

### 4.1. Cell Cultures

The NCI-H295R (ATCC^®^ CRL-2128TM) cell line from ATCC was cultured in Dulbecco’s Modified Eagle Medium (DMEM)/Nutrient mixture F-12 Ham (1:1) supplemented with GlutaMAX-I (Thermo Fisher Scientific31331-028, Thermo Fisher Scientific, Waltham, MA, USA), 2.5% Nu-SerumTM (Corning, 355100, Corning, NY, USA), 1:100 Insulin-Transferrin-Selenium Premix (Corning, 354350, Corning, NY, USA), 100 U/mL penicillin and 100 μg/mL streptomycin (Life Technologies, Thermo Fisher Scientific, Waltham, MA, USA, 15140122). They were seeded at a density of 50,000 cells/cm^2^ for all experiments.

The MDA-MB-468 cell line was cultured in Dulbecco’s Modified Eagles Medium (DMEM) with 4.5 g/L D-glucose, L-glutamine and pyruvate (Thermo Fisher Scientific, 41966-029, Thermo Fisher Scientific, Waltham, MA, USA) supplemented with 10% Fetal Bovine Serum, 100 U/mL penicillin and 100 μg/mL streptomycin (Thermo Fisher Scientific,15140122, Thermo Fisher Scientific, Waltham, MA, USA). They were plated at a density of 20,000 cells/cm^2^ for all experiments.

Cells were cultured in a humidified incubator with 5% CO_2_. 24 h after plating, cells were treated with palbociclib, ribociclib or Dimethyl sulfoxide (DMSO) for 96 h or 48 h for the NCI-H295R and MDA-MB-468 cell lines, respectively. Palbociclib (PD-0332991, A8316) and Ribociclib (LEE-011, A8641), were purchased from CliniSciences (Nanterre, France). Palbociclib and Ribociclib 2 mM stock solutions were prepared in DMSO.

### 4.2. Analysis of Cellular Viability

After treatment, viability was measured using the CellTiter-Glo^®^ Luminescent Cell Viability kit (Promega, G9681, Promega, Madison, WI, USA), following the manufacturer’s instructions. Luminescence was measured with a SpectraMax i3 Multi-Mode Microplate Detection Platform (Molecular Devices, Sunnyvale, CA, USA). Assays were performed in duplicates, in three independent experiments.

### 4.3. Apoptosis Analysis

Apoptosis was measured using the Caspase-Glo 3/7 Assay system (G8090, Promega, Madison, WI, USA) as recommended by the manufacturer. Luminescence was measured with a SpectraMax i3 Multi-Mode Microplate Detection Platform (Molecular Devices, Sunnyvale, CA, USA). Assays were performed in duplicates, in two independent experiments.

### 4.4. Cell Cycle Analyses by Flow Cytometry

Cells were incubated with 10 μM EdU (5-ethynyl-2′-deoxyuridine) in culture media for 90 min before the end of the drug treatment. Click-it reactions were performed using the Click- iT Plus EdU Alexa Fluor 647 Flow CytometryKit (Invitrogen, C10419, Thermo Fisher Scientific, Waltham, MA, USA) according to the manufacturer’s recommendations. The cells were then counterstained with propidium iodide (Invitrogen, P3566, Thermo Fisher Scientific, Waltham, MA, USA) and treated with RNase A (Roche, #10109169001, Roche, Basel, Switzerland) for 30 min. The cell cycle profile was generated using a CyAn ADP 9C analyzer (Beckman-Coulter, Brea, CA, USA). The analysis was performed with the Kaluza software version 2.1 (Beckman-Coulter, Brea, CA, USA). The cell cycle was studied on technical duplicates, in more than three independent experiments.

### 4.5. Total Protein Extraction

Total protein extraction was performed with the same number of cells for each condition. The pellets were incubated in Blue Loading Buffer (New England Biolabs, B7703S, New England Biolabs, Ipswich, MA, USA) with 0.1 unit/mL Benzonase (Merck, 70664-3, Merck, Darmstadt, Germany) for 1 h at room temperature. The extracted proteins were denatured at 95 °C for 5 min.

### 4.6. Western Blot

Proteins were separated on NuPAGE 4–NuPAGE 412% Bis-Tris gels (Invitrogen, NP0341BOX, Thermo Fisher Scientific, Waltham, MA, USA) and transferred onto nitrocellulose membranes (Invitrogen, IB301031, Thermo Fisher Scientifc, Waltham, MA, USA). Primary antibodies (Table 1) were diluted at an adequate concentration in Tris-buffered saline solution with 0.05% TWEEN^®^ 20. Secondary antibodies were used as recommended by the manufacturer. Relative quantifications were performed on three different western blot experiments.

### 4.7. Analysis of the Replication Timing Program

Cells were incubated with 50μM BrdU for 2 h before the end point of treatments, harvested and fixed with cold 70% ethanol. Fixed cells were treated with RNaseA (0.5 mg/mL) and propidium iodide (50 μg/mL) for 30 min at room temperature before cell sorting. 100,000 cells were sorted in two fractions using INFLUX 500 (Cytopeia by BD Bioscience, Franklin Lakes, NJ, USA). Cells were incubated in lysis buffer (50 mM Tris pH = 8; 10 mM EDTA, 300 mM NaCl) and proteinase K (Thermo-Scientific EO-0491, Thermo Fisher Scientific, Waltham, MA, USA) (0.2 mg/mL) for 2 h at 65 °C. DNA was extracted using phenol–chloroform and precipitated with ethanol and sodium acetate (3 M, pH = 5.5; Invitrogen #AM9740, Thermo Fisher Scientific, Waltham, MA, USA). After sonication, DNA was denatured at 95 °C for 5 min and kept on ice for 10 min. Immunoprecipitation of nascent DNA was performed as described in our previous study [21]. Briefly, nascent DNA was immunoprecipitated using the IP-STAR apparatus (Diagenode, Liege, Belgium) with BrdU antibody (10 μg, Anti-BrdU Pure, BD Bioscience, 347580, Franklin Lakes, NJ, USA). Immunoprecipitated nascent DNA was purified and precipitated, as mentioned above.

DNA was amplified using Seq-plex, as mentioned by the manufacturer (Sigma, SEQXE, Merck, Darmstadt, Germany). Each fraction of sorted cells was labeled either Cy3 or Cy5 ULS molecules, respectively (KREATECH, EA-005, Kreatech, Amsterdam, The Netherlands), and as recommended by the manufacturer. The hybridization was performed according to the manufacturer instructions on 4 × 180 K human microarrays (SurePrint G3 Human CGH Microarray Kit, 4 × 180 K, AGILENT Technologies, genome reference Hg18, G4449A, Agilent Technologies, Santa Clara, CA, USA) that cover the whole genome with one probe every 13 kb. Microarrays were scanned with an Agilent’s High-resolution C scanner using a resolution of 3 μm and the autofocus option. Software “START-R” was used for analysis of replication timing [21]. Two independent experiments are performed for each condition.

### 4.8. Molecular Combing and Immunodetection

Cells were pulse-labeled with 50 μM IdU and 50 μM CldU for 30 min consecutively. Then, cells were embedded in agarose plugs (1% low melting agarose, 2-Hydroxyethylagarose, A4018-Sigma) and treated with Proteinase K (1 mg/mL, ROCHE, 3115879001, Roche, Basel, Switzerland) for 48 h. After melting and digestion with β-Agarase (6.25 units/mL, New England Biolabs, M0392L, New England Biolabs, Ipswich, MA, USA), DNA fibers were stretched at a rate of 300 μm/s on silanized coverslips. DNA was denatured with 1 N NaOH solution and immunodetections were performed with the following antibodies: (1) mouse anti-bromodeoxyuridine (BrdU) (BD Biosciences, #347583, BD Biosciences, Franklin Lakes, NJ, USA) and rat anti-BrdU (Abcam, ab6326, Abcam, Cambridge, England), (2) A488-conjugated goat anti-mouse (Invitrogen, A11029, Thermo Fisher Scientifc, Waltham, MA, USA) and A555-conjugated goat anti-rat (Abcam, A21434), (3) mouse anti-single-stranded DNA (Millipore, MAB 3034, Merck Millipore, Burlington, MA, USA), (4) cy5.5-conjugated goat anti-mouse (Abcam, ab6947) and (5) cy5.5-conjugated donkey anti-goat (Abcam, ab6951).

An epifluorescence microscope (Axio Imager Z2; Carl Zeiss, Oberkochen, Germany) equipped with a 63× or 40× objective lens (PL APO, NA 1.4 Oil DIC M27) and a motorized stage (M- 689 XY Microscope Stage with PILine Motor; Physik Instrumente, Karlsruhe, Germany) connected to a charge-coupled device camera (Cool-SNAP HQ2; Roper Scientific, Trenton, NJ, USA) was used. MetaMorph software (Roper Scientific, Trenton, NJ, USA) allowed image acquisition and the study of replication dynamics. Statistical analysis was performed as described in the article [22]. Two independent experiments were performed for each condition.

### 4.9. Fractionation of Chromatin-Bound Proteins

The same number of cells was used for each condition to extract chromatin bound proteins. The extraction was performed by the following steps. Harvested cells were lysed in Buffer A (10 mM Hepes pH 7.9, 10 mM KCl, 1.5 mM MgCl2, 0.34 M sucrose, 10% glycerol, 1 mM DTT, 10 mM NaF, 1X complete EDTA-free Protease Inhibitor Cocktail (Sigma Aldrich, St. Louis, MO, USA) and 0.1% Triton X-100) for 5 min on ice and centrifuged at 1300× *g* for 4 min. The pellet was washed in buffer B (3 mM EDTA, 0.2 mM EGTA, 1 mM DTT, 10 mM NaF, 1X Protease inhibitors) for 10 min on ice and centrifuged at 1700× *g* for 4 min. The pellet was resuspended in buffer C (3 mM EDTA, 0,2 mM EGTA) and centrifuged at 10,000× *g* for 1 min. The pellet, corresponding to the chromatin fraction, was incubated in 1X Blue Loading Buffer (NEB, B7703S), 5 mM MgCl_2_ and 0.1 unit/mL Benzonase (Merck, 70664-3) for 30 min at room temperature. The extracted proteins were then denatured at 95 °C for 5 min.

### 4.10. RT-qPCR

RNAs were extracted with the Nucleospin^®^ RNA extraction kit (Macherey-Nagel, Düren, Germany), following the manufacturer’s instructions. Reverse transcription of RNA was performed with the Transcriptase inverse SuperScript II (Invitrogen, 18064022), following the manufacturer’s instructions. Quantitative PCRs were performed with 1 μL of 1/10 diluted first strands of cDNA in a total volume of 15 μL of 1X KAPA SYBR FAST qPCR Master Mix (Kapa Biosystem, KK4610, Merck, Darmstadt, Germany), containing 10 nM of each PCR primer (Table 2). Reactions were performed in 20 μL LightCycler capillaries (Roche). PCR amplifications were performed with a LightCycler 1.5 thermocycler and analyzed with the LightCycler Software 3.5 (Roche, Basel, Switzerland). Assays were performed in technical duplicates on RNAs extracted from two or three independent experiments.

## Figures and Tables

**Figure 1 ijms-24-10629-f001:**
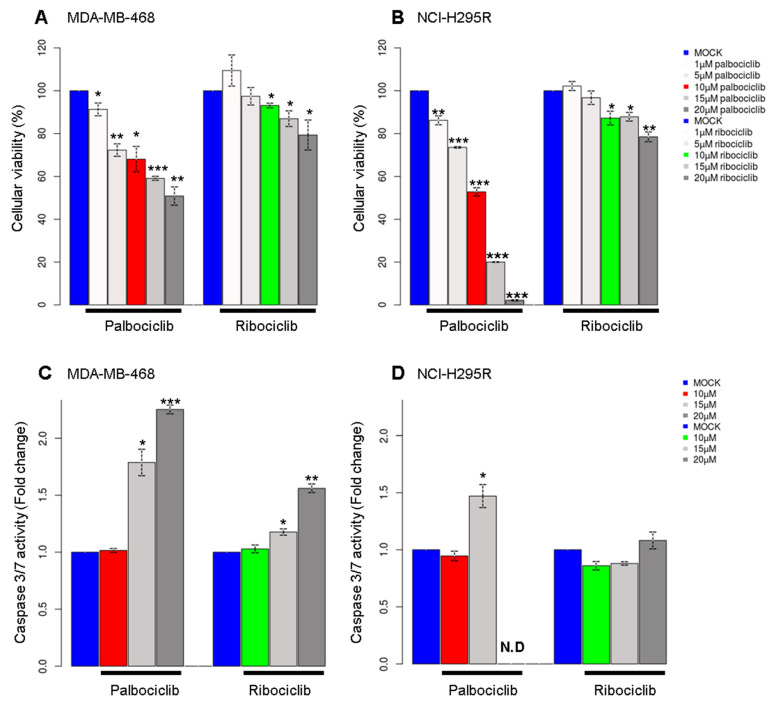
Analysis of cellular viability and apoptosis in pRb-deficient cell lines. (**A**) The viability of cells is measured after 48 h of treatment with increasing doses of palbociclib or ribociclib in MDA-MB-468 cells and (**B**) in NCI-H295R cells. (**C**) Histograms representing the activity of caspase3/7 after treatment with palbociclib or ribociclib in MDA-MB-468 cells and (**D**) in NCI-H295R cells. N.D = Non determined. Two or more independent experiments were performed for each condition (The statistical significance was tested with *t*-test; * *p* < 0.05, ** *p* < 0.01, *** *p* < 0.001).

**Figure 2 ijms-24-10629-f002:**
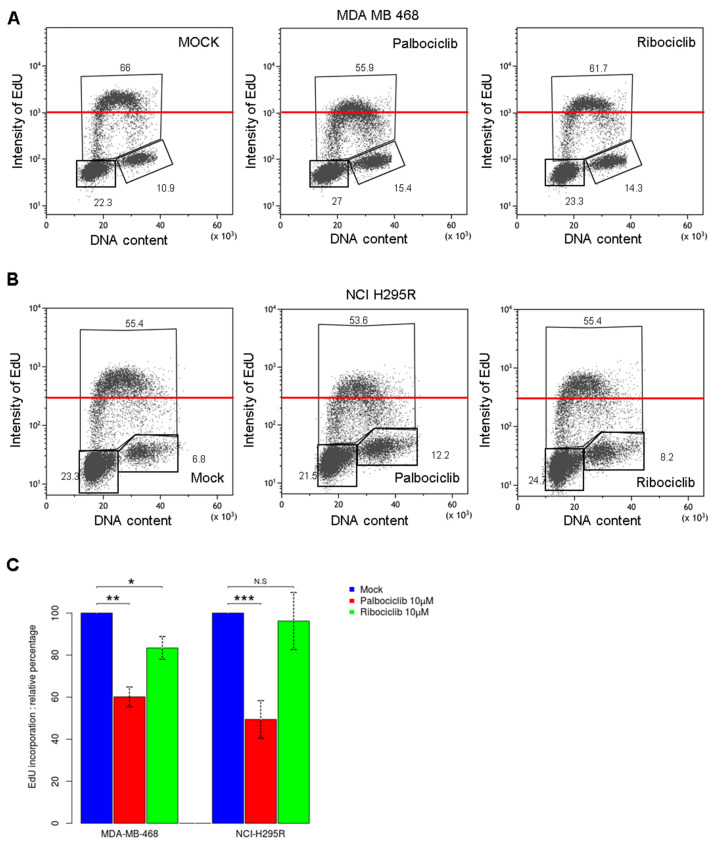
Effects of palbociclib and ribociclib on DNA synthesis during S phase. (**A**) Cell cycle profiles after EdU incorporation and drug treatment in MDA-MB-468 cells and (**B**) in NCI-H295R. The red lines represent the same level of incorporated EdU for each condition. (**C**) Histogram representing the relative percentage of EdU incorporated in S phase in MDA-MB-468 and NCI-H295R cell lines. (The statistical significance was tested with *t*-test; * *p* < 0.05, ** *p* < 0.01, *** *p* < 0.001). (**D**) Total protein levels of Chk1, Chk1 phosphorylated on Serine 345 residues and γH2AX were measured through Western blot after 48 h or 96 h of the treatments in MDA-MB-468 and NCI-H295R cells, respectively. α-Tub and histone H3 were used as loading controls. (**E**) Histograms representing the relative quantity of each protein normalized to the controls. (The statistical significance was tested with *t*-test; * *p* < 0.05, ** *p* < 0.01). Two or more independent experiments were performed for each analysis (**C**,**E**).

**Figure 3 ijms-24-10629-f003:**
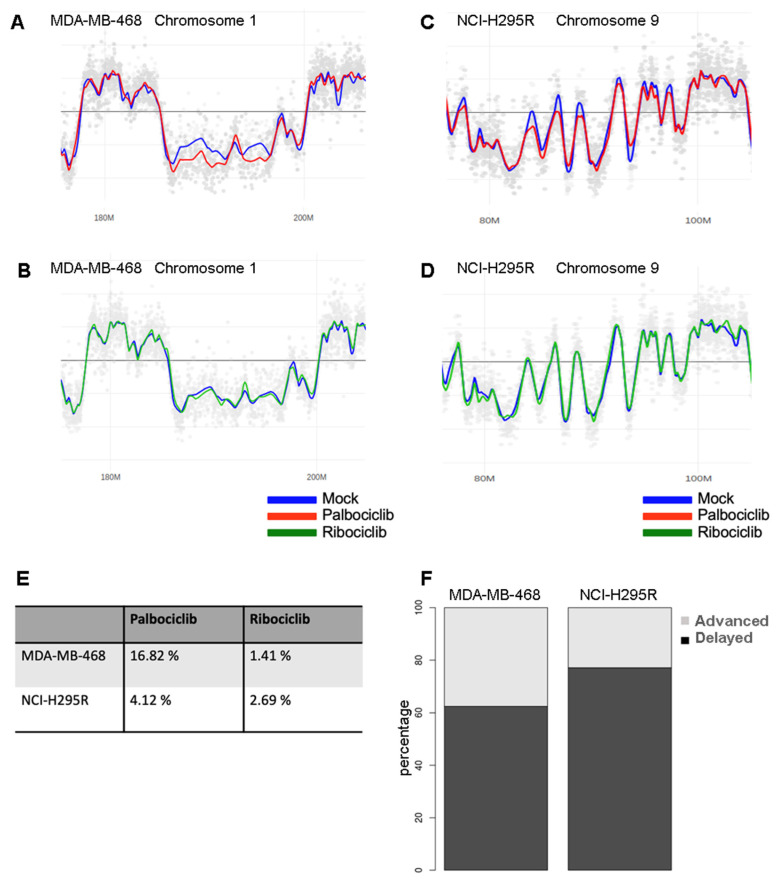
Analysis of the dynamics of DNA replication in MDA-MB-468 and NCI-H295R cells treated with palbociclib. (**A**,**B**) Part of chromosome 1 replication timing profiles in MDA-MB-468 cells. The blue line represents control cells treated with mock, the red one cells treated with palbociclib and the green one with ribociclib. Chromosome coordinates are indicated below each profile (M = Mégabase). (**C**,**D**) Profile of replication timing for a part of chromosome 9 in NCI-H295R cells. (**E**) Percentages of the whole genome altered after the treatment with either palbociclib or ribociclib. (**F**) Stacked histograms representing the proportion of advanced or delayed replicating regions in percentage considering their length after the palbociclib treatment.

**Figure 4 ijms-24-10629-f004:**
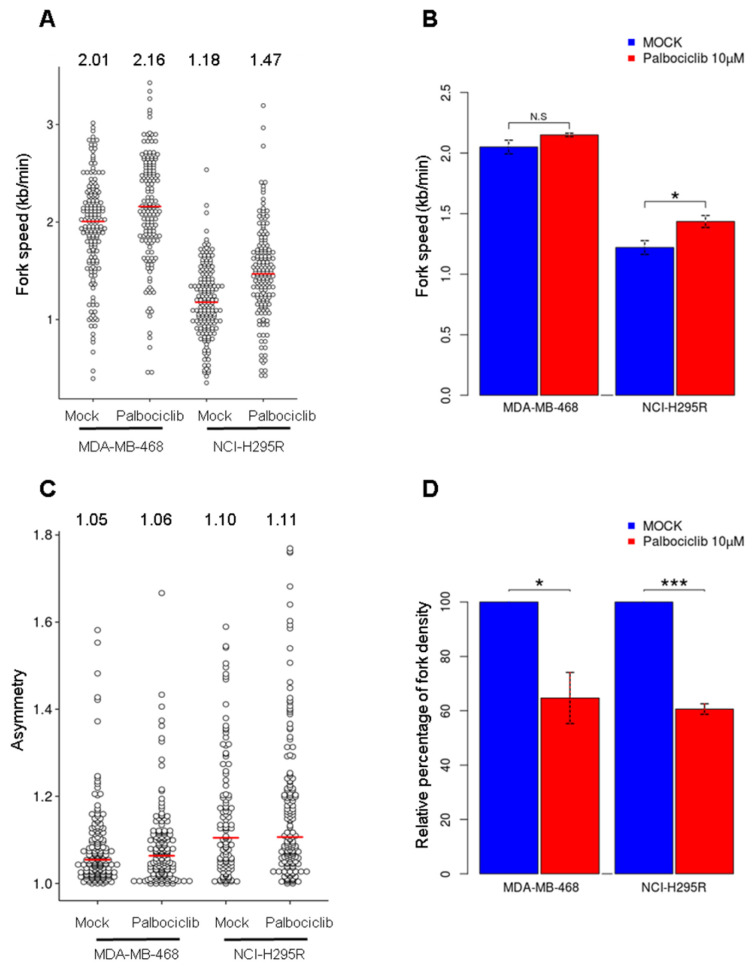
Analysis of the dynamics of DNA replication in MDA-MB-468 and NCI-H295R cells treated with palbociclib. (**A**) Dotplot indicating the fork speed of both cell lines with or without palbociclib treatment. The red lines show the median for each condition. A minimum of 150 forks were scored for each condition. (**B**) Histogram representing the fork speed mean in kb/min for each cell line. The statistical significance was tested with *t*-test; * *p* < 0.05, *** *p* < 0.001) (**C**) Fork asymmetry is measured by CldU/IdU or IdU/CldU ratios higher than 1. The red lines indicate the median of fork asymmetry. (Palbo = palbociclib treatment, Ribo = ribociclib treatment) (**D**) Histogram representing relative percentage of fork density in cells treated with palbociclib compared to control cells treated with mock. The density of forks is calculated as the number of forks divided by the sum of the total length of the fibers. A minimum of 250 kb of fiber length is counted for each condition. The statistical significance was tested as mentioned previously. Two independent experiments were performed for each condition.

**Figure 5 ijms-24-10629-f005:**
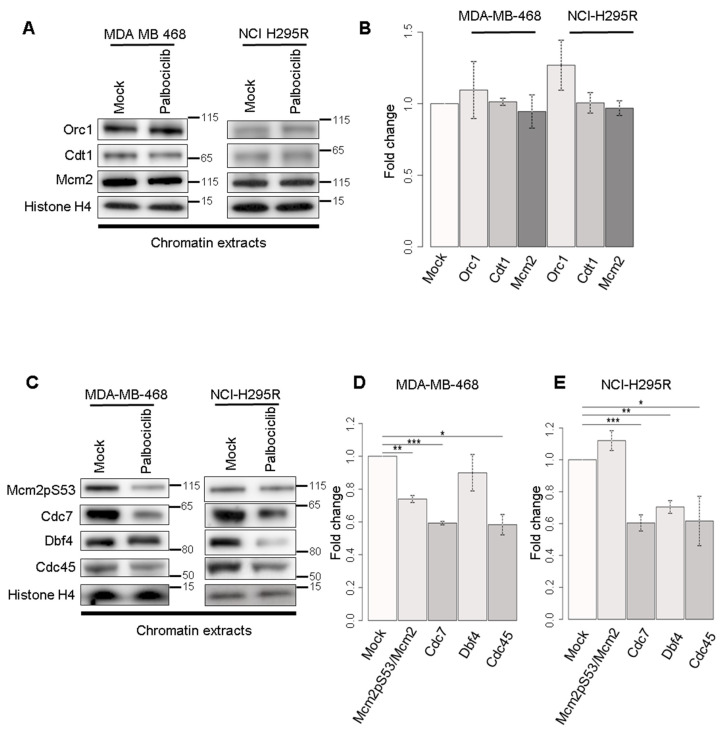
Analysis of chromatin loading of pre-ICs proteins and expression levels of Pre-IC genes in MDA-MB-468 and NCI-H295R cells treated with palbociclib. (**A**) The fraction of proteins of the pre-RC bound to chromatin was analyzed via Western blotting. The same number of cells are used for fractionation at each condition. Histone H4 is a loading control. (**B**) Histogram representing the relative level of pre-RC proteins loaded on chromatin. (**C**) Histogram representing the relative level of pre-RC proteins loaded on chromatin. (**D**) Histogram representing the relative level of pre-IC proteins loaded on chromatin in MDA-MB-468 cells and (**E**) in NCI-H295R cells. Two independent experiments were performed for each condition. (statistical significance; * *p* < 0.05, ** *p* < 0.01, *** *p* < 0.001). (**F**) The total level of pre-initiation complex proteins was analyzed via Western blotting. The same number of cells was loaded for each condition. α-Tubulin is a loading control. (**G**) Histogram representing the relative total amount of proteins in MDA-MB-468 cells and (**H**) in NCI-H295R cells. (**I**) Histograms representing fold change of gene expression level for MDA-MB-468 cells treated with palbociclib for 24 h and (**J**) for NCI-H295R cells treated with palbociclib for 48 h. The level of the TATA-box-binding protein (Tbp) was considered as a control and assessed as 1, from which expression of the other genes was normalized. Two independent experiments were performed for each condition (statistical significance; * *p* < 0.05, ** *p* < 0.01, *** *p* < 0.001).

**Table 1 ijms-24-10629-t001:** Primary antibodies used in this study.

Primary Antibody	Concentration	Reference	Manufacturer
Mouse-anti-Chk1	1/1000	#2360	Cell signaling
Rabbit-anti-Chk1 pS345	1/500	#2348	Cell signaling
Rabbit-anti-γH2AX	1/1000	#9718	Cell signaling
Mouse-anti-αTubulin	1/2000	T9026	Sigma
Rabbit-anti-Orc1	1/500	Ab88530	Abcam
Rabbit-anti-Cdt1	1/250	D10F11	Cell signaling
Rabbit-anti-Mcm2	1/1000	4461	Abcam
Rabbit-anti-Mcm2 phosphoS53	1/2000	Ab109133	Abcam
Mouse-anti-Cdc7	1/1000	Sc 56275	Santa Cruz
Rabbit-anti-Dbf4	1/20,000	Ab 124707	Abcam
Mouse-anti-Cdc45	1/1000	Sc 55569	Santa Cruz
Rabbit-anti-Histone H4	1/2000	07-108	Millipore

**Table 2 ijms-24-10629-t002:** Primers’ list.

Genes	Sequence (3′->5′)
*Tbp*	Forward:cacgaaccacggcactgatt Reverse:ttttcttgctgccagtctggac
*Cdc7*	Forward:caaagtgccccaatcaaact Reverse:tgggccaaagcagttaaatc
*Dbf4*	Forward:aaaccacttcacctcatccc Reverse:tttactccccatgacaaggc
*Cdc45*	Forward:gcaaacacctgctcaagtcc Reverse:tcccaaaaaagttcttcctgtc
*Orc1*	Forward:gccaaagaagagtctcaagcc Reverse:acagcagaaacatgcagcc
*Mcm2*	Forward:cgtatccgaatccaggagag Reverse:gtgttgagggagccatcatag

## Data Availability

DNA replication timing data are available under accession number GSE180530.

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
