# Peer review of "Firing of Replication Origins Is Disturbed by a CDK4/6 Inhibitor in a pRb-Independent Manner"

_ijms, 2023, doi:10.3390/ijms241310629_

Round 1
Reviewer 1 Report
The paper was well written and presented. The paper can be accepted with minor revision
1 Improvement of Figures 5A &5C (better blots) if possible
2. Page alignments for Figure 2D
3. Proofreading the whole manuscript
Dear Editor,
The paper was well written and presented. The paper can be accepted with minor revision
1 Improvement of Figures 5A &5C (better blots)
2. Page alignments for Figure 2D
3. Proofreading the manuscript for English
Author Response
The Reviewer 3 raised some points:
1. We thank the reviewer for bringing to our attention that Figures 5A and 5C needed to be improved.
The reviewer 1 also made this comment (point 2 of reviewer 1). Thus, we improved the quality of the blots in
these figures.
2. As relevantly suggested by the reviewer, we checked the page alignments for Figure 2D.
3. As also wisely suggested by the reviewer, the whole manuscript was proofread by an English native
speake
Reviewer 2 Report
This article examines the influence of CDK4/6 inhibitors plabociclib and ribociclib on cell-cycle distribution and cell death, DNA replication and genome integrity in Rb-deficient cell lines. The authors show that high dose treatments with palbociclib and ribociclib decrease cell viability and induce apoptosis in MDA-MB-486 and NCI-H295R breast and adrenocortical carcinoma cell lines using CellTiter- and Caspase-Glo assays. The effect is much more pronounced with palbociclib compared to ribociclib. They then show that DNA synthesis as measured by EdU incorporation is decreased in palbociclib-treated cells and that replication stress as measured by CHK1 pS345 and g-H2A.X immunoblotting is increased by palbociclib treatment but not by ribociclib. Analysis of replication timing shows that palbociclib but not ribociclib perturbs the replication timing program of up to 16 % of the genome in MDA-MB-468 cells. DNA fiber assays did not show much alteration in fork speed or asymmetry but a significant decrease in fork density was detected by combing assays suggesting defects in origin firing. Chromatin fractionation assays showed that the pre-replication complex components were unaffected by Palbociclib treatment but that some pre-initiation factors had decreased chromatin loading and overall levels in both treated cell lines. qRT-PCR suggested that palbociclib treatment perturbs the transcription of these genes, leading to inefficient production of pre-initiation complex.
The article is generally correctly written, the fact that very high concentrations of the inhibitors are used is acknowledged and is interesting in the sense that clearly off-target effects of these drugs need to be better studied especially since so many clinical trials are ongoing. Perhaps a bit more discussion regarding the recently described off-target effects of these drugs and how these may be interpreted in the context of their own results would make the discussion more interesting (reference 31). There are a number of typos throughout the text that need to be corrected prior to publication. Additionally, some immunoblots are lackluster and would benefit from recropping and adding molecular weight markers to ensure that the bands do correspond to the indicated proteins.
Minor points :
1. Minor/Major point : The link for the supplementary figures did not work. I was unable to assess their quality.
2. Some figures need to be edited : Fig 1A center the Y axis title. In A and B these should be in % viability not fold change. Fig.3 Advanced Fig.4C Asymmetry. The blots in figures 5C are strange, especially cdc7 and Dbf4. The unaltered blot figures are worse… This should be addressed, molecular weight markers and appropriate cropping must be done as well (larger so that we can assess the specificity of the bands). In Fig 5 fold changes should be used in all panels (D,E,G,H, I and J) to make the figure more uniform. In panel F, Palbociclib is placed above the top immunoblotting panels.
3. The sentence on line 550-551 does not make sense.
4. Some of the text is not formatted in the correct font/letter sizes. Uniform fonts/sizes will be required prior to publication.
5. Some of the references are incomplete (14, 29…) revise thoroughly.
Typos/Grammar :
Line 2. Firing of Replication Origins is Disturbed by a CDK4/6 Inhibitor in a pRb-independent manner
Line 14. , underlining a need to characterize the effects of CDK4/6 inhibitors beyond pRb.
Line 30. Dependent
Line 31. highly selective and specific
Line 33. Outcomes
Line 150 cell lines
Line 159 certain hydroxyurea concentration (state the concentration)
Line 174 gH2AX phosphoprotein
Line 180 interferences in DNA synthesis
Figure 2 legend there are 0’s in the text that need to be removed.
Lines 310 observed
Line 386 treated with palbociclib
Line 390 origin firing
Line 467 TATA-box-binding protein
Line 468 assessed as 1
Line 476 were both reduced after 48 hrs of treatment
Line 477 Cdc45
Line 483 levels of
Line 491 is a promising
Line 493 E2F transcription factors that regulate
Line 498 detected little effect
Line 499 dose for pRB-proficient cell lines
Line 525 for fork slowing or collapse
Line 547 How could plabociclib alter the expression of Cdc7 and Cdc45 genes in a pRB-independent manner ?
English is fine but some typos and sentences need to be corrected.
Author Response
The Reviewer 1 raised some points:
1. We thank the reviewer for bringing to our attention that the link for the supplementary figures did not
work. We modified the file with the supplementary figures and uploaded it again.
2. We thank the reviewer for pointing out that some figures needed to be edited. As suggested, we
centered the Y axis title in Fig 1A and mentioned that the Y axis in A and B correspond to cellular viability (%).
We corrected the misspelling of Advanced in Fig 3 and Asymmetry in Fig 4. We improved the quality of the
Western blots in Fig 5C and added the corresponding molecular weights on the Western blots in Fig 2D, 5A,
5C and 5F and also in the supplementary Fig 4. We also modified all the graphs in Fig 5 to have fold changes
in the Y axis of all D, E, F, G, H and I panels to make the figure more uniform. In the 5F panel, we placed the
Palbociclib name in a spot where it no longer overlapped the immunoblot. The original blots file (in pdf format)
is following this letter and will be added as an extra-file in the IJMS site.
3. We agreed with the reviewer that the sentence initially on line 550-551 and now on line 390-391 of
the revised document did not make sense. We modified this sentence and also the previous sentence to make
it clearer. The text is now the following:
«The regulation of expression of numerous genes has been studied in pRb-deficient cell lines under a
treatment with 1µM palbociclib as a negative control of their expression in a pRb-efficient cell line. However,
no conclusion was drawn concerning the detected gene expression variations in pRb-deficient cells [28,29].»
4. The reviewer 1 and the reviewer 2 noticed that some part of the text is not formatted correctly. As
suggested, we checked that the whole document was formatted with the appropriate fonts and letter sizes as
required.
5. We thank the reviewer for bringing to our attention that some references were incomplete,
particularly the 14th and the 29th. We thoroughly revised all the references and placed them in the appropriate
format and in bracket [14] along the text.
As wisely suggested by the reviewer, we made all the typography and grammar corrections throughout
the text (initially from line 2 to 547, now from line 2 to 387 of the revised document). We indicated them by
adding “changes recommended by reviewer1” as commentaries in the manuscript.
We also precised the concentration of hydroxyurea used (50 and 100 mM, initially on line 159, now on line 126
of the manuscript).
The reviewer also pointed out that we could discuss a little more about the off-target effects of the
drugs we used in this study and how it could be interpreted in the context of pRB-deficient cells in link with the
reference 31. Thus, we added the following sentence to the text (originally on line 556 and now on line 397-
401 of the revised document):
«Multi-omic analysis have shown that palbociclib inhibits CDK2/cyclinA1 and CDK2/cylinE1 in the µM
range [12]. However, a study revealed that 10 µM of palbociclib, the concentration of palbociclib that we used
in pRb-deficient cells, has a target occupancy of 65% for CDK2 [11], making it sufficient to target CDK2.»

Reviewer 3 Report
Title:
The replication origins firing is disturbed by a CDK4/6 in inhibitor in a pRb independent manner
Authors
Su-Jung KIM, etc.
Summarized the viewpoint
The authors investigated the effective mechanism of CDK4/6 inhibitors on pRb-deficient cancer cells. Palbociclib impaired the initiation of DNA replication by impairing the assembly of the pre-IC onto chromatin during the S-phase. The reduction of Cdc7 and Cdc45 transcripts in pRB-deficient cancer cell lines treated with Palbociclib was determined. However, cell lines treated with Ribociclib, another DCK4/6 inhibitor, had not been shown. Authors concluded that Palbociclib is a promising cancer drug for cancer cells, including pRb-deficient cancer cells. The manuscript needs some minor changes for publication.
Detailed
Please use the same font to entire manuscript.
Authors summarized the possibility of using palbociclib as a combination therapy in the abstract. How about drug toxicity to patients when used as a combination therapy for pRb-positive cancer? How to find and what are the candidate drugs to combine with Palbociclib to enhance anticancer efficacy against pRb-deficient cancer cells? Please discuss in the discussion.
Figure 2D: What did phosph-gH2AX data look like as an indicator of DNA damage?
Figure 3F: What was the unit of Y-axis?
Author Response
The Reviewer 2 raised some points:
The reviewer wondered about the drug toxicity of palbociclib to patients when used as a combination
therapy for pRb-positive cancer.
We had some sentences in the discussion on line 328-332 of the revised document : “ In pRb-positive
cancer, neutropenia seems to be the predominant effect of a palbociclib treatment with an associated
tolerance depending on individuals. Today, palbociclib treatment is also associated with Endocrine Therapy.
On these long-term safety analyses, there is no evidence of specific cumulative or delayed toxicities [41].”
Concerning which candidate drug should be combined with palbociclib in pRb-deficient cancer cells
and, since we are not medical specialists of cancer treatments, we added the following sentence to the
conclusion of the manuscript on line 437-444 of the revised document :
«A recent paper from Knudsen's team shows that pRb-deficient cells, which are resistant to specific
chemotherapeutic agents such as carboplatin, are highly sensitive to PLK1 and CHK1 inhibitors. These
inhibitors reduce pRb-deficient cells survival [42]. The same study also shows a significant effect on pRbdeficient xenografts. Finally, the authors suggest that this type of inhibitors could be used in pRb-deficient
triple-negative breast cancer. From this point of view, we suggest that palbociclib may be used for pRbdeficient tumors combined with other therapeutic molecules inhibiting PLK1 and CHK1, such as Volasertib and
AZD7762, respectively.»
The reviewer also wondered if the phospho-gH2AX data are indicative of DNA damage in Figure 2D.
We are confident that our data is an indicator of DNA damage as we observed a comparable or even higher
increase in the gH2AX signal after palbociclib treatment in MDA-MB-468 and NCI-H295R cells than the one
induced by 1mM hydroxyurea treatment. This dose of hydroxurea is well-known to induce DNA damage in
human cells (for example, treatment used as a control in Snyder, A. R.; et al. Therapeutic doses of
hydroxyurea cause telomere dysfunction and reduce TRF2 binding to telomeres. Cancer biology & therapy,
2009, 8, 1136–1145.)
We thank the reviewer for noticing that the Y axis unit of the Figure 3F was missing. So, we added
“percentage” as the unit of the Y axis in Figure 3 and in its legend and added a related commentary in the text.